# A versatile genetic engineering toolkit for *E. coli* based on CRISPR-prime editing

Yaojun Tong [1,2✉], Tue S. Jørgensen[1], Christopher M. Whitford [1], Tilmann Weber [1✉] & Sang Yup Lee [1,3✉]

CRISPR base editing is a powerful method to engineer bacterial genomes. However, it restricts editing to single-nucleotide substitutions. Here, to address this challenge, we adapt a CRISPR-Prime Editing-based, DSB-free, versatile, and single-nucleotide resolution genetic manipulation toolkit for prokaryotes. It can introduce substitutions, deletions, insertions, and the combination thereof, both in plasmids and the chromosome of *E. coli* with high fidelity. Notably, under optimal conditions, the efficiency of 1-bp deletions reach up to 40%. Moreover, deletions of up to 97 bp and insertions up to 33 bp were successful with the toolkit in *E. coli*, however, efficiencies dropped sharply with increased fragment sizes. With a second guide RNA, our toolkit can achieve multiplexed editing albeit with low efficiency. Here we report not only a useful addition to the genome engineering arsenal for *E. coli*, but also a potential basis for the development of similar toolkits for other bacteria.

[1] The Novo Nordisk Foundation Center for Biosustainability, Technical University of Denmark, Lyngby, Denmark. [2] State Key Laboratory of Microbial Metabolism, School of Life Sciences and Biotechnology, Shanghai Jiao Tong University (SJTU), Shanghai, China. [3] Department of Chemical and Biomolecular Engineering, BioProcess Engineering Research Center, BioInformatics Research Center, Institute for the BioCentury, Korea Advanced Institute of Science and Technology (KAIST), Daejeon, Republic of Korea. ✉email: yaojun.tong@sjtu.edu.cn; tiwe@biosustain.dtu.dk; leesy@kaist.ac.kr

Advances in synthetic biology, metabolic engineering, multiomics, high throughput DNA sequencing and synthesis, and computational biology have prompted a rapidly increasing demand for fast and robust genetic engineering methods to speed up the strain development in a Design-Build-Test-Learn cycle. The classic genetic engineering approaches in prokaryotes often use phage-derived RecET and lambda red recombinase-based recombineering[1,2]. They employ the homology-directed integration/replacement of a donor double stranded DNA (dsDNA) or oligonucleotide for making insertions, deletions, and substitutions of the target DNA. For example, the "Multiplex Automated Genome Engineering" (MAGE)[3] is a method that can be used for simultaneous manipulation of genes across multiple chromosomal loci of *E. coli*. Possible mutations include mismatch mutation, insertion, and deletion, and editing efficiencies are usually below 20% for all types of edits[3]. Classical MAGE not only requires the synthesis and delivery of ssDNA oligos but also the expression of lambda (λ) red recombinase systems (Exo, Beta, and Gam) in the target *E. coli* strain[3]. Several improved methods have been developed based on the classical MAGE to increase the editing efficiency and decrease the off-target effect. For example, the pORTMAGE system[4], using a dominant-negative mutant protein of the MMR pathway, not only achieves higher editing efficiency and lower off-target effect, but also works for different bacterial species other than *E. coli*. One step forward, an improved pORTMAGE system was built by discovery of new, highly active single-stranded DNA-annealing proteins (SSAP). The identified CspRec improved pORTMAGE editing efficiency to up to 50%[5]. Recently, retron library recombineering was introduced as a new method that achieves up to 90% editing efficiency by in vivo production of single-stranded DNA using the targeted reverse-transcription activity of retrons, however, such editing efficiencies require disrupting multiple DNA repair pathways in the host cell[6], which heavily limits its applications.

CRISPR-Cas (Clustered regularly interspaced short palindromic repeats-CRISPR associated (Cas) proteins) systems, originating from the bacterial adaptive immune system[7], have been engineered as genome editing tools for a variety of organisms[8]. Among these tools, the Class 2, type II CRISPR system CRISPR-Cas9 of *Streptococcus pyogenes* has been most widely studied and applied. The Cas9 nuclease can be guided by an engineered RNA (single guide RNA, sgRNA) to make DNA double strand breaks (DSBs) of the protospacer adjacent motif (PAM)-containing target DNA[9]. Different types of genetic engineering can be achieved during the repair of DSBs. There are two major pathways for DSB repair in vivo, the non-homologous end joining (NHEJ) and the homology-directed repair (HDR)[10]. In most eukaryotes, NHEJ is the dominant way to repair DSBs. During NHEJ repair, small insertions and/or deletions (indels) are introduced at the lesion site, leading to gene disruptions in the target gene. In most bacteria, DSBs normally lead to cell death due to the lack of NHEJ[11]. In these organisms, DNA damage is primarily repaired via HDR with sister chromatids[11], where template DNA replace the damaged DNA fragment by recombination[12].

The lack of NHEJ repair in most prokaryotes restricts the direct use of CRISPR-Cas9 without providing editing templates as a genome editing tool. However, the method is widely used for negative selection to eliminate wild-type cells in recombination-based engineering methods[13]. Unlike CRISPR-Cas9, "DSB-free" CRISPR base editing systems have successfully been applied for direct genome editing in a number of bacteria without providing editing templates[14–16]. As they rely on DNA deaminase reactions, CRISPR base editors can only make one type of changes to the DNA: the substitution (C to T/A/G, or A to G), and the target C

or A has to be within the relatively narrow editing window. Hence, it soon becomes a bottleneck of applying CRISPR base editors for bacterial genome engineering. For the insertion of large DNA fragments, methods such as CRISPR-associated transposase (CAST)[17] and INsert Transposable Elements by Guide RNA-Assisted TargEting (INTEGRATE)[18] were developed by combining CRISPR-Cas systems and transposons. The INTEGRATE was successfully tested in *E. coli* for integrating a ~10.1 kb fragment into the chromosome[18].

Recently, reverse transcriptase-Cas9 H840A nickase (Cas9n)-mediated targeted prime editing (PE) has been demonstrated in human cells[19], rice and wheat cells[20] to directly knock-out, knock-in, and replace nucleotides at the target locus without introducing DSBs and requiring editing templates. The CRISPR-PE system uses the 3′-extension sequence of the modified sgRNA (herein named as PEgRNA) to provide a primer binding sequence (PBS) and a reverse transcription template (RTT) carrying the desired edits for reverse transcription with the reverse transcriptase that is allocated in the target locus by Cas9n:sgRNA. After DNA repair, designed mutations are introduced into the target locus. As the system only introduces a nick in one DNA strand, we hypothesized that it may not cause cell death in bacteria and thus could be applicable in bacterial genome engineering as well. Here, we report the establishment and evaluation of the CRISPR Prime Editing toolkit for *E. coli*.

## Results

**Design of CRISPR-prime editing system for *E. coli*.** To evaluate if the reverse transcriptase-Cas9n-mediated DNA modification works in bacteria, we constructed a three-plasmid system (pCDF-GFPplus, pPEgRNA, and pCRISPR-PE). A fourth plasmid (pVRb_PEgRNA, Supplementary Fig. 5) is introduced for the multiplexed editing. Plasmid pCDF-GFPplus serves as the reporter plasmid harboring a gene encoding an *E. coli* codon optimized fast folding GFP[21] under a constitutive promoter J23106 (Fig. 1a). Plasmid pPEgRNA carries the constitutive promoter J23119 driving PEgRNA transcription. The PEgRNA is composed of a 20-nt spacer and a 3′ extension containing the PBS and RTT (Fig. 1b). The third plasmid pCRISPR-PE expresses an *E. coli* codon optimized fusion protein composed of an engineered reverse transcriptase M-MLV2 (moloney murine leukemia virus variant[19]), a flexible linker, and a Cas9n (Cas9 nickase, the H840A mutant of SpyCas9) under a tetracycline-inducible promoter (Fig. 1c).

**Validation of CRISPR-prime editing system on plasmid editing in *E. coli*.** To assess the versatility of the CRISPR-Prime Editing system on plasmid DNA engineering in *E. coli*, we designed a full set of possible DNA engineering events, including insertions, deletions, substitutions, and combinations of these to introduce premature stop codons into the coding sequence of GFP. The loss of fluorescence enables easy screening and evaluation for desired editing events. We identified a protospacer located at positions 178–197 of the GFP coding sequence (Supplementary Table 3) that should allow the introduction of a stop codon by DNA engineering with designed PEgRNA. The testing was initiated following observations reported in human cells[19] with a 3′-extension consisting of 13 nt PBS and 13 nt RTT scaffold. In the case of insertion, the length of RTT equals the RTT scaffold size plus the designed insertion, for example the length of RTT for TAA insertion is 16 nt (Supplementary Table 3 and Supplementary Table 4). Designed edits were placed inside the potential editing window starting from the nick and continuing downstream[19] (Fig. 1a). The Cas9n-M-MLV2 fusion protein binds to the desired PEgRNA transcript, forming an

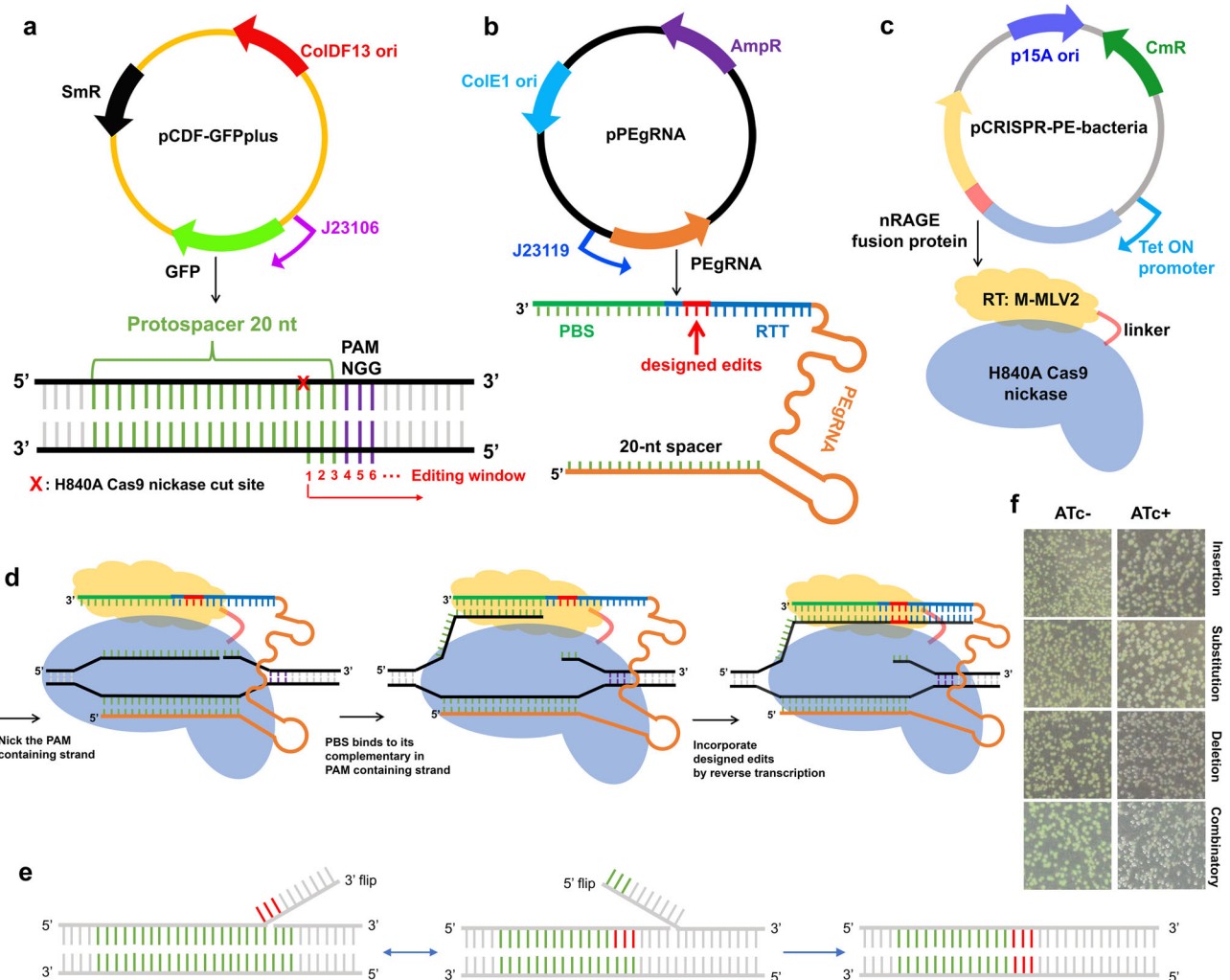

**Fig. 1 A three-plasmid system for evaluation of CRISPR-Prime Editing system in *E. coli*. a** The plasmid map of the reporter vector pCDF-GFPplus, which carries a constitutive promoter J23106 driving expression of fast folding GFP. The plasmid contains a spectinomycin-resistance (SmR) gene, and the ColDF13 origin (ori). An illustration of the target DNA composition is shown below the plasmid map. **b** The plasmid map of the PEgRNA transcript bearing vector, which contains the ColE1 ori and an ampicillin-resistance (AmpR) gene for selection. The PEgRNA transcript is under control of the constitutive promoter J23119. A detailed structure is shown beneath, the 3′ extension sequence is composed of a PBS (in green) and a RTT (in blue), which carries the intended edits (in red). **c** Plasmid map of the CRISPR-PE-bacteria vector, which carries a p15A ori and a chloramphenicol-resistance gene (CmR) for selection. The *E. coli* codon optimized, tetracycline inducible promoter driven Cas9n-M-MLV2 fusion protein consists of a H840A Cas9 nickase (Cas9n), a 33-aa flexible linker, and a moloney murine leukemia virus (M-MLV) variant M-MLV2, described previously[19] with the following mutations: D200N, L603W, T306K, W313F, and T330P compared to the WT M-MLV (GenBank: AAC82568.2). **d** A schematic model for DNA engineering with CRISPR-Prime Editing system for *E. coli*. After being expressed, the Cas9n-M-MLV2:PEgRNA complex binds to the targeted DNA sequence in a sgRNA-dependent and PAM-dependent manner. The Cas9n domain within the fusion protein nicks the PAM-containing strand, freeing the adjacent DNA sequence. Subsequently, this piece of single stranded DNA hybridizes to the PBS, then primes reverse transcription of new DNA containing the designed edits based on the RTT within the 3′-extension of the PEgRNA transcript. **e** Two possible consequences of CRISPR-Prime Editing. It normally has an equilibration between the edited 3′ flap and the unedited 5′ flap, only the cleavage of the 5′ flap leads to the desired editing. **f** Colony views of *E. coli* strains transformed with CRISPR-Prime Editing systems carrying designed edits of TAA insertion, T to A substitution, T deletion and the combinatorial edits with and without 200 ng/mL ATc induction using a Doc-It imaging station, non-green colonies appeared after 24 h induction of 200 ng/mL ATc.

RNA-protein complex, the Cas9n-component of the complex subsequently finds its target DNA sequence and introduces a nick in the PAM containing DNA strand. The PBS within the 3′ extension then binds to the flipped PAM containing DNA sequence, initiating the reverse transcription to elongate the nicked DNA sequence based on the sequence of the RTT (Fig. 1d). After the reverse transcription process, the nicked double stranded DNA hypothetically undergoes an equilibration between the edited 3′ flap and the unedited 5′ flap. The cleavage of the unedited 5′ flap then leads to the desired DNA editing[19] (Fig. 1e).

As a proof of concept, we transformed *E. coli* cells with CRISPR-Prime Editing systems programmed for TAA (3-bp) insertion, T to A (1-bp) substitution, T (1-bp) deletion, and the combination thereof. All of these edits will lead to stop codons that prematurely terminate translation to inactivate the target gene (Supplementary Table 4). Growth profiling of *E. coli* strains bearing different plasmids with and without induction indicated that almost no changes in fitness were caused by non-targeting CRISPR-Prime Editing system (Supplementary Table 5). However, a mild negative effect of around 10% on growth (maximal doubling time) due to the 200 ng/mL anhydrotetracycline (ATc)

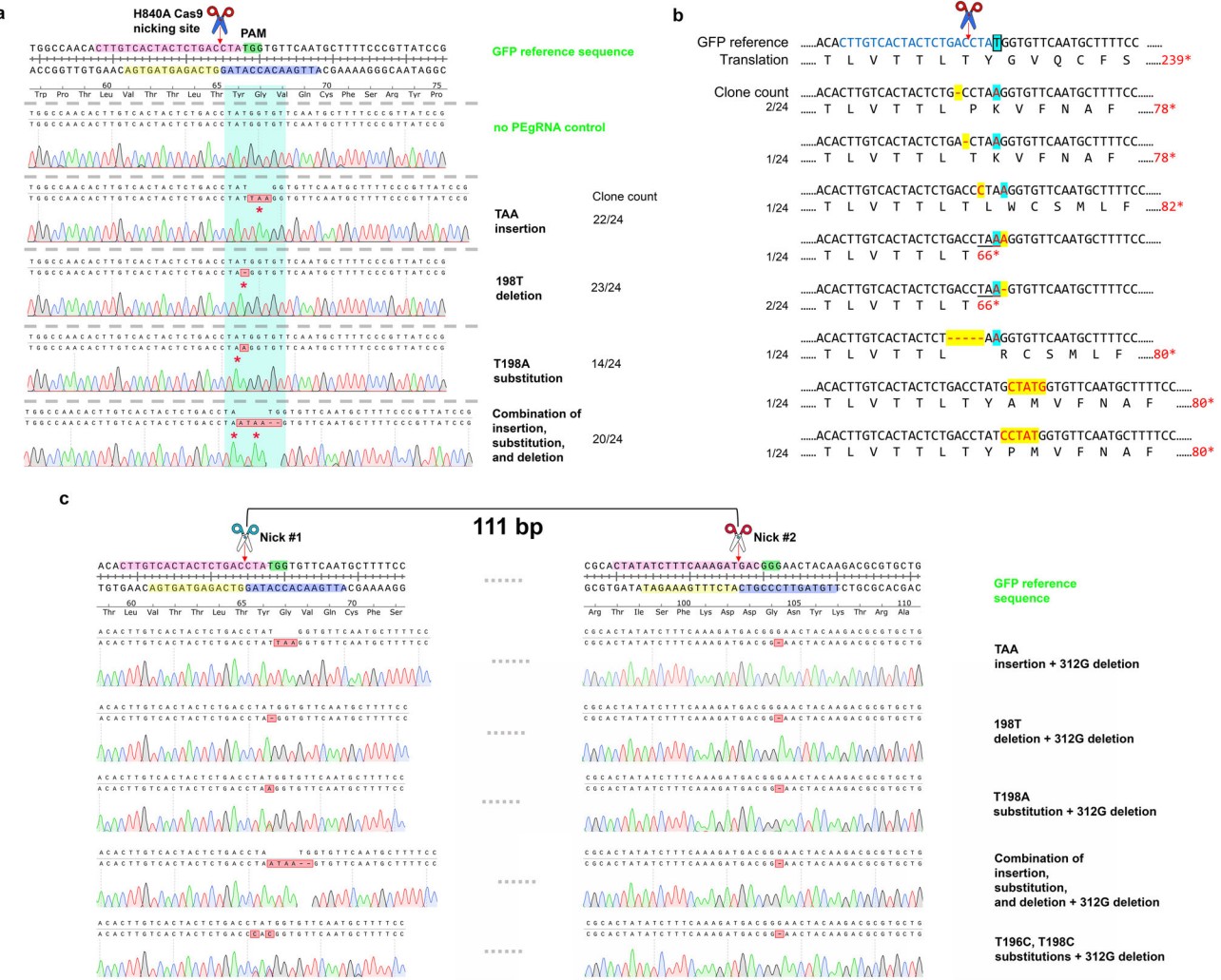

**Fig. 2 Evaluation of plasmid-based editing by CRISPR-Prime Editing system using Sanger sequencing.** Stop codon is displayed as an asterisk under the nucleotide sequences, the potential Cas9 H840A nicking site is indicated by a red arrow. The 20 nt protospacer, the PAM sequence, the PBS, and the RTT are highlighted in pink, green, yellow, and blue, respectively; while the cyan masked Sanger sequencing traces show the sequence to be replaced by the RTT that will contain the different edits designed. **a** Twenty-four randomly picked colonies of each designed DNA engineering were Sanger sequenced and traces were aligned to the targeted locus of the GFP coding sequence. The correctly edited clone numbers and the total sequenced clone numbers are shown on the right of the figure. **b** Shown are the recorded target-specific unintended edits of the 1 bp substitution. The target T is boxed and masked in light blue. The GFP reference DNA sequence and the translated amino acid sequences are show on the top row. The corrected edited nucleotide is in red and masked with light blue, while the unexpected mutations (off-target) are in red and masked with yellow. The recorded off-target clone numbers and the total sequenced clone numbers are shown on the left of the figure. **c** Sanger sequencing traces of the successfully dual-edited clones by CRISPR-Prime Editing. Two nicks are 111 bp away, the left nick (nick #1) is introduced by pPEgRNA (ColE1 ori), and the right nick (nick #2) is introduced by pVRb_PEgRNA (pSC101 ori). The combinations of 3 bp insertion, 1 bp deletion, 1 bp substitution, 2 bp substitution, combinatory editing, and 1 bp deletion are displayed.

induction was observed in a clonal formation unit (CFU) assay using solid agar plates (Supplementary Table 5), and a liquid cultivation assay with 96-well microtiter plates (Supplementary Fig. 6 and Supplementary Table 6). After induction with 200 ng/mL ATc, we observed non-fluorescent (non-green) clones formed on all four plates of designed DNA editing events (Fig. 1f). In order to further confirm that GFP fluorescence loss was due to the designed DNA editing consequences, we randomly Sanger sequenced 24 non-fluorescent colonies from each induced plate. Results demonstrated that almost all of the non-fluorescent colonies were indeed carrying the designed stop codon edits (Fig. 2a). By extending the incubation time of the induction plates (for example to 3–5 days), we observed that the editing events were accumulating over time. This becomes visible when colonies from 24 h of incubation are further incubated: non-green "sections" grow out of the original green colony, even

surrounding it (Supplementary Fig. 1). Sanger sequencing confirmed that these no longer fluorescent strains were successfully edited. This result indicates that prolongation of the incubation time is one possibility to increase the number of correctly edited cells.

In the 24 Sanger sequenced clones, we noticed that eight clones harboring indels near the target nucleotide of the T to A substitution editing event (Fig. 2b, A genome-wide off-target evaluation will be presented below), while no such indels were found in other types of editing. We classified this as target adjacent unintended edits, which only occur in physical proximity to the nick site. Besides the combinatorial editing of insertion, deletion, and substitution, we also investigated the possibility of performing double substitutions from one construct. An edit replacing tyrosine at the position 66 of the GFP to histidine (Y66H) was designed by flipping the TAT codon to a

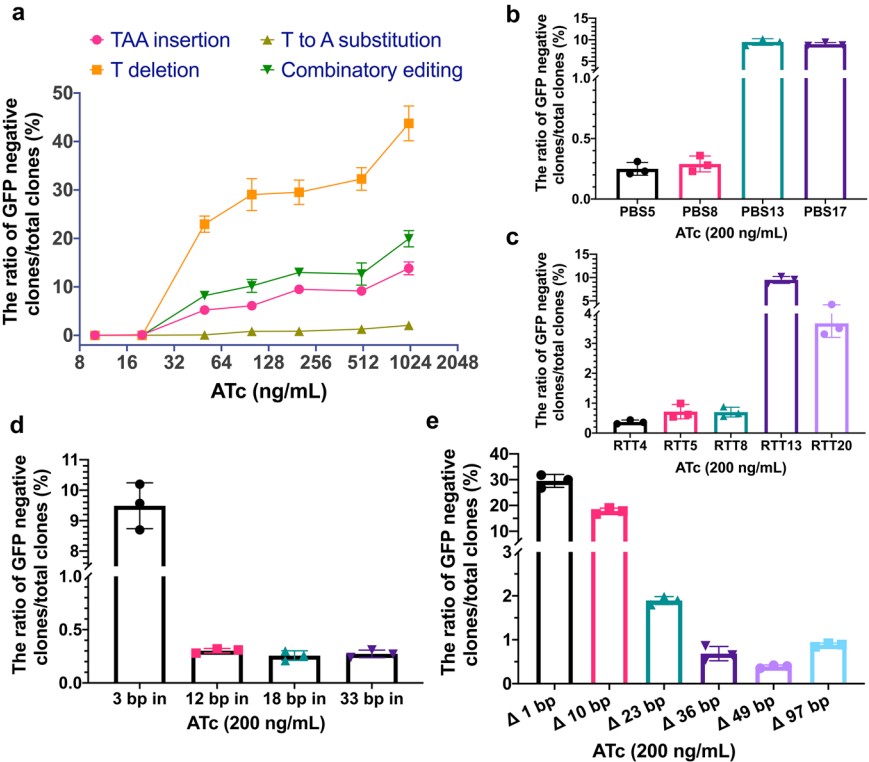

**Fig. 3 Characteristics of CRISPR-Prime Editing system for DNA engineering in *E. coli*.** The editing efficiency was defined as ratio of white clones (GFP-negative)/ total clones on a screening plate. **a.** Eight different concentrations of ATc, ranging from 0 to 1000 ng/mL (0, 10, 20, 50, 100, 200, 500, and 1000) were used to evaluate the induction of CRISPR-Prime Editing system on four DNA engineering events of 3 bp insertion, 1 bp deletion, 1 bp substitution, and the combinatorial editing. **b** The evaluation of PBS length. **c** The evaluation of RTT scaffold length. **d** The capacity of DNA fragment insertion with different sizes. **e** The capacity of DNA fragment deletion with different sizes. Mean ± s.d. of three biological replicates are shown. 200 ng/mL of ATc was used for **b–e**.

CAC codon. We obtained eight non-fluorescent clones out of which six clones were correctly edited (Supplementary Fig. 2). To further assess whether CRISPR-Prime Editing system for *E. coli* is capable of multiplexed editing, we introduced another compatible plasmid pVRb carrying a second PEgRNA for a G deletion (pVRb_PEgRNA_312Gdel) into *E. coli* DH10B harboring the 3-plasmid system for TAA insertion, T deletion, T to an A substitution, two Ts to two Cs substitution, and combinatory editing. We successfully identified all expected dual-editing events (Fig. 2c), however the editing efficiency was relatively low (<1%).

**Characterization of CRISPR-prime editing system in *E. coli*.** As the Cas9n-M-MLV2 fusion protein is driven by the ATc inducible promoter, we evaluated the optimal condition of induction using eight different ATc concentrations. The editing efficiencies were defined by calculating the ratio of non-green colonies. We observed a dose dependent induction manner for all four designed DNA engineering events (Fig. 3a). For cases of 1 bp deletion, 3 bp insertion, 1 bp substitution, and the combinatorial editing, CRISPR-Prime Editing system can reach efficiencies up to 43.7%, 13.8%, 19.9%, and 2.1%, respectively with 1000 ng/mL of inducer (Fig. 3a). Deletion and insertion of similar sized DNA fragments has an efficiency equal to, or higher than, MAGE[3]. Among the four DNA editing events, deletion showed the highest, while substitution showed the lowest efficiency (Fig. 3a). The editing efficiency did not change significantly with ATc concentrations of 100–500 ng/mL, and thus we decided to induce CRISPR-Prime Editing system with 200 ng/mL ATc in the following experiments unless specified otherwise.

Next, we evaluated the optimal length of PBS and RTT by measuring the frequency of the TAA-STOP codon insertion. Nearly no edits were observed when a PBS of 5 nt or 8 nt was used, while a 17 nt PBS showed editing efficiency equivalent to a 13 nt PBS (Fig. 3b). This indicated that the window of PBS for CRISPR-Prime Editing system for *E. coli* is 13 nt to 17 nt. For the RTT scaffold, we designed five different lengths, 4, 5, 8, 13, and 20 nt. We observed that too short (like <10 nt) or too long (like >20 nt) RTT reduced the editing efficiency, and the optimal length of the RTT scaffold was around 13 nt (Fig. 3c, d). Results obtained in this study are consistent with previous reports in eukaryotes[19,20] Moving forward, by using the optimal length of PBS and RTT scaffold, we systematically tested the capacity of both insertion and deletion with 200 ng/mL of inducer. We designed insertions of 3, 12, 18, and 33 bp, in which the 18 bp fragment is the mini-T7 promoter; and deletions of 1, 10, 23, 36, 49, and 97 bp. Though clones with all designed DNA engineering events could be successfully obtained, the editing efficiency dropped greatly with the increase of sizes (Fig. 3d, e). For instance, under 200 ng/mL ATc, the editing efficiencies of 1 bp deletion and 10 bp deletion can reach 29.5% and 17.8%, respectively, while efficiencies dropped to below 2% with lengths of 23–97 bp. For insertion, the efficiency was in general lower than deletion. The efficiency of 3 bp insertion was about 10% with 200 ng/mL ATc, and it dropped to below 1% when the size increased to 18–33 bp (Figs. 3d, e and Supplementary Fig. 3). In several of these cases, the editing efficiency was low. Many clones carrying the activated CRISPR-Prime Editing systems still showed GFP fluorescence. We randomly picked 10 of these "escapers", together with four controls (Supplementary Table 1). The 14 strains were sequenced, and analyzed with our genome-wide

SNP profiling approach that was used for the off/on-target evaluation as well. Seven out of ten "escapers" lost the 26 bp 3′ extension sequence (Supplementary Table 7); except these deletions, the other parts of plasmids and the chromosome were intact. In 3 "escapers", no mutations/SNPs were identified both on plasmids and chromosome that can explain why no CRISPR-Prime Editing occurred (Supplementary Table 7). This indicates that besides mutating the guide RNA, yet-unknown escaping mechanisms are also present in *E. coli*.

Inspired by the observation that a second nick in the non-edited strand would increase the editing efficiency of CRISPR-Prime Editing in some mammalian[19] and plant cells[20], we designed and validated two strategies of the second nick introduction in *E. coli* (Supplementary Note 1). There were almost no visible colonies after the second nick was introduced (Supplementary Fig. 4). This result indicates that in *E. coli*, which does not have a NHEJ pathway, introducing the second nick cannot increase the editing efficiency but only compromises the use of CRISPR-Prime Editing system.

**Assessing the ability of chromosomal DNA editing with CRISPR-Prime Editing system in *E. coli*.** Beyond editing plasmid DNA, we also assessed if CRISPR-Prime Editing system is capable of engineering chromosomal DNA in *E. coli*. To this end, two metabolic pathways for lactose and D-galactose degradation in *E. coli* MG1655 were selected. β-galactosidase, encoded by the *lacZ* gene within the lactose metabolic pathway, metabolizes X-gal (5-bromo-4-chloro-3-indolyl-β-D-galactoside, an analog of lactose) into 5-bromo-4-chloro-indoxyl, which will form dark blue 5,5′-dibromo-4,4′-dichloro-indigo by oxidation. On the contrary, X-gal remains colorless if the *lacZ* gene is inactivated (Fig. 4a, b). An early stop codon was designed to be introduced into the *lacZ* gene of *E. coli* MG1655 by insertion of TAG, deletion of GC, and substitution of GT to TA. In general, the editing efficiencies similar to those for plasmid DNA engineering were observed by counting the white colonies out of the total formed colonies on X-gal supplemented LB plates (Fig. 4c). Editing efficiencies of substitution, insertion, and deletion were 6.8%, 12.2%, and 26%, respectively. To validate the editing events, the targeted region of eight randomly picked non-blue colonies from each designed DNA engineering event were PCR amplified and further subjected to Sanger sequencing. All sequenced clones bore the expected edits (Fig. 4d–f). Moreover, another gene, the *galK* gene from the Leloir pathway of D-galactose metabolism in *E. coli* MG1655 was tested. The loss of function of the *galK* gene can be positively selected by supplementing a galactose analog 2-deoxy-D-galactose (2-DOG), as 2-DOG will be metabolized by galactokinase (encoded by the *galK* gene) to form a toxic compound 2-deoxy-galactose-1-phosphate, which cannot be further metabolized[22] (Fig. 4g, h). A TAA stop codon was designed to be inserted into the *galK* gene in *E. coli* MG1655 strain. Of the visible colonies on the 2-DOG supplemented M63 agar plate, all four that were randomly picked showed the expected insertion (Fig. 4i).

**Genome-wide on/off-target evaluation of the CRISPR-Prime Editing system in *E. coli*.** Precision is one of the most important requirements for DNA engineering. We applied a bacterial genome-wide SNP (Single Nucleotide Polymorphism) profiling approach[21] to systematically evaluate the on-target and potential off-target mutations caused by the CRISPR-Prime Editing system. We selected one clone of each designed DNA engineering events of both plasmid- and chromosome-based editing (Table 1 and Table S1) for on/off target mutation analysis using whole-genome sequencing. In order to assess the background noise of mutations,

we also sequenced the two parental strains DH10B and MG1655 that were used in this work (Table S1). As expected, all designed editing events (on-target mutations) were present in the corresponding strains (Table 1). For potential off-target mutation analysis, we examined mutations on both, plasmids and chromosome, using breseq[23]. Only one single nucleotide substitution in the chromosome was identified in the edited strains of 1 bp deletion, 1 bp substitution, and the combinatorial editing (Table 1). No off-target mutations were found in the 3 bp insertion and 2 bp substitution edited strains (Table 1). As we observed some target specific off-target mutations in the editing event of 1 bp substitution (Fig. 2b), we wanted to further investigate if long-distance off-target mutations would be introduced in the target specific off-target mutation carrying strain. To this end, another clone (DH10B-plasmids-PE_1bpsub) of 1 bp substitution and a Sanger sequencing recorded deletion of 5 bp upstream of this designed substitution was subjected to whole-genome sequencing analysis (Table 1). Except the expected on-target 1 bp substitution and the target specific off-target mutation, no additional off-target mutations were found (Table 1).

For the chromosome DNA editing events, we also observed a high-fidelity of CRISPR-Prime Editing system. Except the excision of the *insB1–insA* fragment (a mobile element) in some strains, only one single nucleotide substitution was found in the chromosome of the designed 3 bp insertion clone (Table 1). As it has been reported that MG1655 strain will lose the 776 bp insB1–insA fragment during cultivation[24], we therefore excluded this from the CRISPR-Prime Editing system related off-target effect, marking it as one of the parental variable mutations (Table 1).

In summary, our results indicate a very high fidelity of CRISPR-Prime Editing system on both plasmid and chromosome DNA engineering in *E. coli*.

## Discussion

Due to the importance of *E. coli* in basic microbiological studies and biotechnological applications as a workhorse for the production of various bioproducts, there has been continued demand for novel and efficient DNA engineering tools. Widely used and versatile methods for genetic engineering of *E. coli* are RedET and lamda red-based recombineering[1,2], or MAGE-based approaches[3–5]. Although much effort has been exerted to simplify and improve the recombineering protocol, it is either still relatively difficult to operate[1,25], and it requires the target strain to have a specific genetic background, for example the deficiency of methyl-directed mismatch repair or RecA, the key enzyme for recombinational DNA repair[26]. The emergence of CRISPR-based genetic engineering has been revolutionizing biotechnology, however much less applications were reported in prokaryotes than in eukaryotes[8], partially because of the different dominant DSB repair pathways. As a result, in many bacteria, including *E. coli*, CRISPR-Cas9 has been widely employed as a tool for counter-selection to eliminate non-modified cells from a mixed population in homology-directed recombination methods such as the lambda red recombination systems[27]. It remains very challenging to engineer DNA at a single nucleotide level, even when combined with a powerful counter-selection system such as CRISPR-Cas9, the efficiency of making point mutations using oligonucleotide-directed mutagenesis is very low (<3% before optimization)[28].

Recently, two types of CRISPR base editors were developed, which are capable of C to T conversion (CBE) by the cytosine deaminase (APOBEC1 or Target-AID)[29,30], C to G or A substitution with engineered cytosine deaminases[31], and A to G conversion (ABE) by the adenosine deaminase (TadA)[15] without

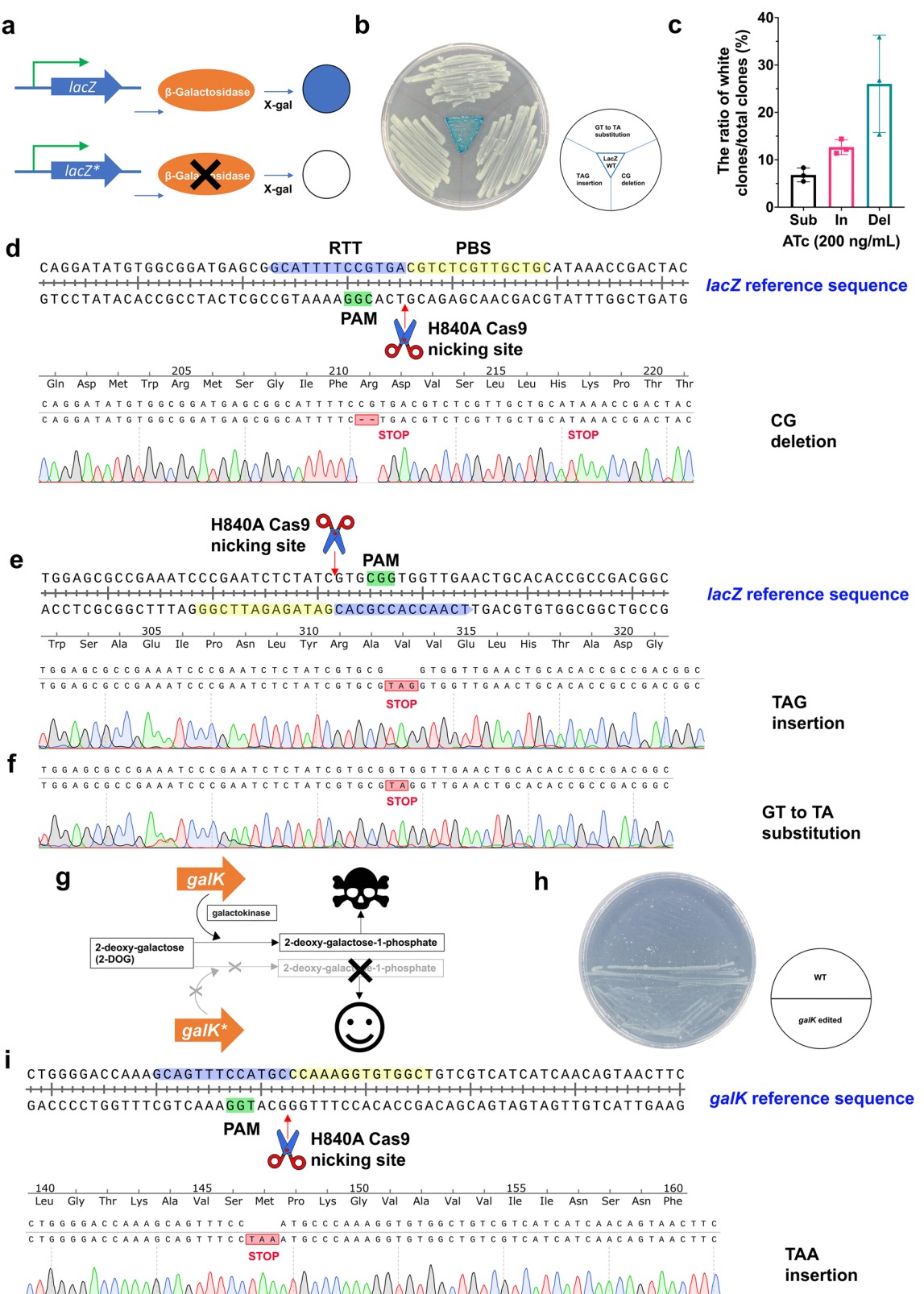

involving DSBs. Thus, they can be directly applied in bacteria for DNA manipulation. One of the main applications is gene inactivation using CBE to convert Arg, Gln, or Trp codons to a stop codon[14,16]. There are also a few cases of using adenosine deaminase based base editing for in vivo protein engineering[14,15]. However, so far, CRISPR base editing technology has not been as widely used in bacteria as expected due to the restriction of fixed

substitutions (C to T/G/A, or A to G) and the relatively narrow editing window (5–7 nucleotides).

We demonstrated in this study that CRISPR-Prime Editing system cannot only make substitutions but also insertion, deletion, and combinatorial editing at single base pair resolution in E. coli without requiring DSBs, editing templates, or homologous recombination. We observed a very high fidelity of using the

**Fig. 4 CRISPR-Prime Editing for *E. coli* is capable of chromosomal DNA editing. a** A graphic illustration of the function of *lacZ* gene. The star within the gene box represents a stop codon being introduced. **b** Three clones of *E. coli* MG1655, where the inactivation of lacZ was confirmed by Sanger sequencing and a wild type *E. coli* MG1655 were re-streaked on an agar plate with X-gal. **c** A bar chart shows the editing efficiencies of chromosomal DNA engineering by 3 bp insertion, 2 bp deletion and 2-bp substitution by calculating the ratio of white clones/total clones on an induction LB plate with X-gal supplemented. Mean and standard deviation of three biological replicates are shown. **d–f** Eight non-blue colonies were picked and Sanger sequenced. Sequencing traces aligned with non-edited *lacZ* reference sequences are displayed. Shown are alignments of 2 bp deletion (**d**); 3 bp (stop codon TAG) insertion (**e**); and 2-bp substitution (**f**). **g** A graphic illustration of the function of *galK* gene. **h** An agar plate view of the successful *galK* gene inactivation in *E. coli* MG1655 strains by CRISPR-Prime Editing. **i** Four colonies from the 2-DOG supplemented plate were picked and Sanger sequenced. Sequencing traces aligned with non-edited *galK* reference sequences are displayed. The alignment of 3 bp (stop codon TAA) insertion is shown. The potential Cas9 H840A nicking site is indicated by a red arrow. The 20 nt protospacer, the PAM sequence, the PBS, and the RTT are highlighted in pink, green, yellow, and blue, respectively. The translated amino acid sequences together with the introduced stop codons are labeled underneath each nucleotide sequence. Numbers in red on the right side show the correct and total Sanger sequenced colonies.

system in *E. coli*, while the mutation rates/off-target effects in the MAGE and other recombineering systems are much higher, and normally it requires pre-engineering of the host strains when using these systems[3–5]. CRISPR-Prime Editing system for *E. coli* has great potential in expanding the possibilities of DNA engineering, although further studies are required to further increase its editing efficiency. As a result of its high modularity and simple composition, CRISPR-Prime Editing for *E. coli* might be multiplexed by providing a PEgRNA self-processing machinery like Csy4[14] and consequently applied for high-throughput mutagenesis applications. However, it has to be noted that the editing efficiency was extremely low in our proof-of-concept multiplexing approach using the strategy of providing two PEgRNA delivery plasmids. CRISPR-Prime Editing also has the potential of being applicable to a wider range of bacteria including those previously considered difficult to be genetically engineered. As in the case of CRISPR-Cas9 and CRISPR base editing systems, the use of other Cas proteins and protein engineering will likely improve editing capabilities by expanding the selection of accepted PAMs and increasing efficiencies[32–35]. Different reverse transcriptases other than the M-MLV could also provide different features to increase the performance of CRISPR prime editing systems in applied organisms. Modulating intracellular DNA repair systems and better designed PEgRNAs could also be helpful in increasing the editing efficiency.

CRISPR-Prime Editing, a versatile DNA engineering system reported in this study, represents a powerful addition to the toolbox of genetic and metabolic engineers not only for *E. coli*, but other organisms. These tools are likely to substantially advance our understanding of basic life science and to increase capabilities for advanced microbial engineering for biotechnological purposes.

## Methods

**Stains, plasmids, media, and growth condition**. All *Escherichia coli* strain and plasmids used in this study are listed in Supplementary Table 1. *E. coli* cultures were grown at 37 °C in LB (both broth and solid) (Sigma, USA). Appropriate antibiotics were supplemented with the following working concentrations: spectinomycin (50 μg/mL), carbenicillin or ampicillin (100 μg/mL), chloramphenicol (25 μg/mL), kanamycin (50 μg/mL), and anhydrotetracycline (0–1 μg/mL). M63 minimal medium was used for positive selection of *galK* mutants. It is composed of 2 g/L $(NH_4)_2SO_4$, 13.6 g/L $KH_2PO_4$, 0.5 mg/L $FeSO_4 \cdot 7H_2O$, 1 mM $MgSO_4$, 0.1 mM $CaCl_2$, and 10 μg/mL thiamine, 0.2% glycerol and 0.1% 2-deoxy-D-galactose. Two percent agar was supplemented when making agar plates. X-gal (5-bromo-4-chloro-3-indolyl-beta-D-galactopyranoside) was used for screening *lacZ* mutants. Prior to use, each LB plate with appropriate antibiotics is plated with 40 μL of 20 mg/mL X-gal. All chemicals involved in this study were from Sigma, USA.

**General protocol of DNA manipulation**. All primers, important sequences, spacers, and 3′ extensions used in this study are listed in Supplementary Tables 2, 3, and 4, respectively. Standard protocols were used for DNA (plasmids and genomic DNA) purification, PCR, and cloning. PCR was performed using Q5® High-Fidelity 2× Master Mix (New England Biolabs, USA). The point mutation in dCas9 to

create H840A Cas9n was made using Q5® Site-Directed Mutagenesis Kit (New England Biolabs, USA). DNA assembly was done by using NEBuilder® HiFi DNA Assembly Master Mix (New England Biolabs, USA) unless specified otherwise. DNA digestion was performed with FastDigest restriction enzymes (Thermo Fisher Scientific, USA) unless specified otherwise. NucleoSpin® Gel and PCR Clean-up kit (Macherey-Nagel, Germany) was used for DNA clean-up both from PCR products and agarose gel extracts. NucleoSpin® Plasmid EasyPure Kit (Macherey-Nagel, Germany) was used for plasmid preparation. Sanger sequencing was carried out using Mix2Seq kit (Eurofins Scientific, Luxembourg). DNA fragments were synthesized by Genscript while oligonucleotides were synthesized by IDT (Integrated DNA Technologies, USA).

All kits and enzymes were used according to the manufacturers' recommendations. We diligently followed all waste disposal regulations of our institute, university, and local government when disposing of waste materials.

**Multiplasmid system design and plasmid construction**. All plasmids constructed in this study have been deposited to Addgene, individual Addgene plasmid number are listed below. Plasmids in the same testing system should be compatible with each other, and therefore they must have different origins of replication (ori). For this purpose, a combination of p15A ori, ColE1 ori, and ColDF13 ori was used.

Synthetic constitutive promoters J23119 (BBa_J23119) and J23106 (BBa_J23106), and the ribosome binding site (RBS) BBa_B0034 were obtained from the registry for standard biological parts in the iGEM Parts Registry (http://parts.igem.org/Main_Page).

The construction of GFP-based reporter plasmid: The plasmid was designed in silico to carry the GFP expression cassette, which is composed of a constitutive promoter J23106, a RBS BBa_B0034, a fast folding GFP variant GFP+[21], and a terminator T0. The GFP+ coding sequence was codon optimized to *E. coli*. The whole cassette was synthesized by Genscript and assembled into the pCDF-1b plasmid (ColDF13 ori, Millipore, USA) replacing the MCS region by Gibson Assembly, and ended up with the plasmid pCDF-GFPplus (Addgene #172718).

The construction of CRISPR-Prime Editing plasmid: Firstly, we created pCas9n(H840A) from pdCas9-bacteria (p15A ori, Addgene plasmid #44249)[36] by site-specific mutation of 10A of dCas9 to 10D using Q5® Site-Directed Mutagenesis Kit (New England Biolabs, USA). Secondly, we designed the 33a linker-M-MLV2 cassette in silico. Linker sequence: SGGSSGGSSGGSETPGTSESATPESSGGSSGGSS. M-MLV2, a moloney murine leukemia virus (M-MLV) variant from a previous study[19] with the following mutations D200N, L603W, T306K, W313F, and T330P compared to the WT M-MLV (GenBank: AAC82568.2). Thirdly, the cassette was codon optimized to *E. coli*, synthesized by Genscript, and then assembled into pCas9n(H840A) to replace the stop codon of Cas9n by Gibson Assembly, resulting in the plasmid pCRISPR-PE-bacteria (Addgene #172715). The fusion protein (cargo) Cas9n-linker-M-MLV2 is under control by a tetracycline inducible promoter.

The construction of PEgRNA transcript carrying plasmid: The empty PEgRNA plasmid was modified from the pgRNA-bacteria (ColE1 ori, Addgene plasmid #44251)[36] by removing the 20 bp spacer, named as pPEgRNA (Addgene #172716). For construction of functional pPEgRNA there were three steps: firstly, a spacer and 3′ extension were designed in silico; secondly, amplification of the functional PEGgRNA cassette using the pPEgRNA as a template was performed concurrently with amplifying the PEgRNA backbone fragment using the primer set (PEgRNA backbone_F and PEgRNA backbone_R); lastly, the functional PEgRNA cassette was assembled into the PEgRNA backbone. Sanger sequencing was used for validation. Spacers and 3′ extensions were designed both manually and using PrimeDesign[37].

For introducing the second nick, we constructed pnsgRNA (pSC101 ori, kanR) by replacing the sfGFP expression cassette in pVRb20_992 (Addgene plasmid #49714)[38] with the sgRNA transcript cassette from pPEgRNA. We first amplified the plasmid backbone of pVRb20_992 and the sgRNA cassette with primer sets of pVRb_backbone_F and pVRb_backbone_R, and sgRNA_cassette_F and sgRNA_cassette_R from pVRb20_992 and pPEgRNA, respectively. Then these two fragments were Gibson assembled and later validated by Sanger sequencing,

**Table 1 Whole-genome sequencing-based analysis of on-target and off-target mutations.**

| Strain | Parental variable mutations | Recorded mutations | | | | Off-target (Plasmids and/or chromsome) |
|---|---|---|---|---|---|---|
| | | On-target | | References (coding strand) | Alleles | |
| | | Genes | Regions in the target gene | | | |
| DH10B-plasmids-PE_3bpin | 0 | GFP in plasmid: pCDF-GFPplus | 283 | - | +TAA | 0 |
| DH10B-plasmids-PE_1bpdel | 0 | GFP in plasmid: pCDF-GFPplus | 283 | T | - | 1 (chr. Pos. 396,221. C→T) |
| DH10B-plasmids-PE_1bpsub | 0 | GFP in plasmid: pCDF-GFPplus | 283 | T | A | 1 (pCDF-GFPplus pos. 278. -A) |
| DH10B-plasmids-PE_1bpsub-2 | 0 | GFP in plasmid: pCDF-GFPplus | 283 | T | A | 1 (chr. Pos. 2,198,023. A→G) |
| DH10B-plasmids-PE_2bpsub | 0 | GFP in plasmid: pCDF-GFPplus | 281–283 | TAT | CAC | 0 |
| DH10B-plasmids-PE_combo | 0 | GFP in plasmid: pCDF-GFPplus | 283–284 | TG | ATTA (T→A, +TAA, −G) | 1 (chr. Pos. 4,047,371. G→T) |
| MG1655-chr-PE_3bpin | 1 (chr. Pos. 1,976,527. Δ776 bp, insB1-insA) | lacZ in chromosome | 364,594 | - | +TAG | 1 (chr. pos. 1,285,157. A→G) |
| MG1655-chr-PE_3bpin-2 | 1 (chr. Pos. 1,976,527. Δ776 bp, insB1-insA) | galK in chromosome | 788,602 | - | +TAA | 0 |
| MG1655-chr-PE_2bpdel | 0 | lacZ in chromosome | 364,898–364,899 | CG | - | 0 |
| MG1655-chr-PE_2bpsub | 1 (chr. Pos. 1,976,527. Δ776 bp, insB1-insA) | lacZ in chromosome | 364,592–364,593 | GT | TA | 0 |

resulting in pnsgRNA plasmid (Addgene #172717). Spacers for introducing the second nick in the nsgRNA paired with the related PEgRNA were designed using PrimeDesign[37]. This plasmid is also used for delivery of the second PEgRNA.

**High throughput electroporation of multiple plasmids.** In vivo assay of strains carrying multiple plasmids were performed from freshly transformed E. coli DH10β strains. A HT Nucleofector™ System (Lonza, Switzerland) together with 96-well Nucleocuvette plates (Lonza, Switzerland) were used for high throughput electroporation. Before electroporation, the 96-well Nucleocuvette plate was transferred from −20 °C to ice for 10 min. Twenty microliter of electrocompetent DH10β or MG1655 E. coli cells with 10% glycerol were added into each desired well, 0.5 μL of each plasmid (about 30 ng) was subsequently added. A total amount of plasmid DNA of <100 ng per transformation normally performed well. The program used in this study is X_bacteria_14, with the code GN-100. After electroporation, 180 μL of fresh LB broth were added into each well. The cultures were then transferred into a 96-deep well plate containing 200 μL of fresh LB broth (making the transformation culture in total 400 μL) for recovery for 1 h at 37 °C and 300 rpm.

**Illumina deep sequencing-based genome-wide on/off-target evaluation and analysis of "escapers".** For on/off-target evaluation, one or two Sanger sequencing validated clones of each designed editing events were selected; while for escapers examination, ten clones with induced CRISPR-Prime Editing systems targeting pCDF-GFPplus, still showing GFP-fluorescence, were randomly picked. Together with necessary control strains, they were inoculated in a 50 mL tube (Greiner Bio-One, Germany) containing 10 mL LB broth without any antibiotics. After incubating at 37 °C, 300 rpm in an INNOVA 44R incubator shaker (Eppendorf, Germany) for 24 h, 5 mL of the culture was used for genomic DNA plus intracellular plasmid DNA isolation with a Blood & Cell Culture DNA mini Kit (Cat No./ID: 13323, Qiagen, Germany). While a NucleoSpin® Plasmid Easy-Pure Kit (Macherey-Nagel, cat. no. 740727.250) was used for the WT pCDF-GFP plasmid isolation. The genomic library construction and illumina paired-end sequencing were carried out by Novogene Co., Ltd. (Beijing, China), using the NEB Next® Ultra™ DNA Library Prep Kit (New England Biolabs, USA) with a target insert size of 350 nt and six PCR cycles.

The illumina reads obtained from the sequenced samples were trimmed using Trim Galore (v. 0.6.4_dev, Cutadapt v. 2.10) with the switches --length 100 and --quality 20. All mutation calls were performed using breseq (v. 0.33.2, bowtie2 v. 2.3.4.1)[23,39] with default parameters. For plasmid-based editing, the E. coli DH10B genome sequence NC_010473 is used as the reference, while for chromosome-based editing, the E. coli MG1655 genome sequence NC_ U00096 is used as the reference, both along with the relevant plasmids. Mutation calls that existed in all samples as well as the parental strain were not counted as off-target effects.

**Editing efficiency evaluation using a fluorescence-based colony counting assay.** Fifty microliter electroporation culture (400 μL in the cases of a second nick is introduced) of each strain was plated onto appropriate antibiotics containing LB agar plates supplemented with and without inducer, respectively. All plates were covered by aluminium foil and incubated at 37 °C for 24 h. After cultivation, total colonies were counted by a Doc-It imaging station (Fisher Scientific, USA) with a trisection protocol. Non-fluorescent colonies in each zone of all three zones were further counted with and without a Blue-Light Transilluminator (Safe Imager 2.0, Thermo Fisher Scientific, USA). The editing efficiency was calculated as: the number of non-green colonies in each zone/total number of visible colonies in the same zone. The graphs are generated by Prism (version 8).

**Editing events confirmation by Sanger sequencing.** Eight to 24 primarily identified positive clones of each strain were picked, and inoculated into 5 mL LB broth with proper antibiotics. After overnight (~16 h) cultivation, cultures were subjected to plasmid isolation using the NucleoSpin® Plasmid EasyPure Kit (Macherey-Nagel, Germany) or colony PCR using Q5® High-Fidelity 2× Master Mix (New England Biolabs, USA) if a chromosomal region was targeted. The isolated plasmids and the cleaned PCR products were Sanger sequenced using the Mix2Seq kit (Eurofins Scientific, Luxembourg) with proper primers. The obtained sequence traces were analyzed and visualized using SnapGene (GSL Biotech, USA).

**Colony forming unit (CFU) assay.** All CFU experiments were performed using E. coli DH10beta electrocompetent cells. The plasmids of interest were measured using a NanoDrop UV spectrophotometer (NanoDrop2000; Thermo Fisher Scientific, USA) and diluted to 100 ng/μL. For each transformation, 100 ng of each plasmid were used and added to 50 μL of electrocompetent cells. The cells were then incubated on ice for 30 min and subsequently electroporated using 1 mm cuvettes and a MicroPulser electroporator (Bio-Rad, USA) using the Ec1 program. Four hundred microliter of LB medium were subsequently added and the cells were incubated in a thermoblock at 37 °C while shaking for 1 h. The transformations were then plated in appropriate dilutions on LB plates supplemented with the corresponding antibiotics with and without ATc. After incubation overnight at 37 °C, the colonies were counted and the real CFUs were calculated by multiplying the counted CFUs with the corresponding dilution factor.

**Growth profiling in liquid culture.** All growth experiments were performed in a ELx808 plate reader (Buch and Holm A/S), set to 37 °C, constant shaking, and measurement of OD630 every 20 min. Measurements were taken for 24 h. All cultivations were performed in 96-well microtiter plates with F-bottom and a lid. For inoculation of the precultures, four colonies were picked from each non-induced LB plate (12 in total) and used to inoculate wells of a 96 deep well plate filled with 1 mL of LB supplemented with the corresponding antibiotics (spectinomycin, 50 μg/mL ampicillin, 100 μg/mL; chloramphenicol, 25 μg/mL; and ATc, 200 ng/mL). The preculture was incubated overnight at 37 °C at 250 rpm. The next morning, the OD630 of the preculture was measured using the ELx808 plate reader. The cultures were diluted 1:2 to obtain values below 0.8. For each well, the necessary volume for inoculation of 200 μL of microtiter cultures with a starting OD630 of 0.05 was calculated. The corresponding volumes were then added to one microtiter plate with 200 μL and the corresponding antibiotics only, and one with both antibiotics and ATc. Both cultivations were run in ELx808 plate readers in parallel using the same run protocol.

**$\mu_{max}$ and the maximum doubling time calculation.** For determination of $\mu_{max}$ and the maximum doubling time based on the Monod equation, the natural logarithm of the OD630 values was plotted against the time in hours. In this plot, the exponential phase can be easily determined based on the linear progression. For each cultivation, the exponential phase was determined using this graphical approach, and linear regressions were calculated for those time points. The slope of the linear regression corresponds to $\mu_{max}$ in $h^{-1}$, which can subsequently be used to calculate the maximum doubling time by dividing $\ln(2)$ by the determined $\mu_{max}$ value.

**Reporting summary.** Further information on research design is available in the Nature Research Reporting Summary linked to this article.

## Data availability
The Illumina re-sequencing data generated in this study have been deposited to NCBI (For a full list of SRA accession numbers, please see Supplementary Table 8). Data for on/off target evaluation: NCBI BioProject PRJNA752926; SRA accessions SRR15371474-SRR15371494. Data for editing escapers: NCBI BioProject PRJNA752927; SRA accessions SRR15371770-SRR15371774. Source data are provided with this paper.

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

## Acknowledgements
We thank Simon Shaw for proofreading the manuscript. We thank Alexandra Hoffmeyer for excellent technical support in Illumina sequencing. This work was supported by grants from the Novo Nordisk Foundation (NNF20CC0035580, NNF15OC0016226, NNF16OC0021746). S.Y.L. was also supported by the Technology Development Program to Solve Climate Changes on Systems Metabolic Engineering for Biorefineries (NRF-2012M1A2A2026556 and NRF-2012M1A2A2026557) from the Ministry of Science and ICT through the National Research Foundation (NRF) of Korea.

## Author contributions
Y.T., T.W. and S.Y.L., conceived and designed the project. Y.T. and C.M.W. carried out the laboratory experiments and analyzed the data. T.S.J. performed computational analysis. Y.T., T.W. and S.Y.L. wrote the manuscript with input from all authors.

## Competing interests
The authors declare no competing interests.
