## [Peer Review File · Nature Communications]

Reviewers' Comments:

Reviewer #1:

Remarks to the Author:

Tong et al report the adaptation of a remarkable technology called Prime Editing, for the first time in bacterial cells. The transfer of such a complex genetic editing platform in bacteria was not trivial, given that it requires the expression of a massive ~238 kDa fusion protein, and it relies on the cell endogenous DNA repair systems to incorporate the edit. The system developed by Tong et al, named CRISPRnRAGE, is surprisingly efficient, and represents a useful addition to the precise genetic editing toolkit in *E. coli*.

In particular, Tong et al. thoroughly investigated how the CRISPRnRAGE technique performs for different types of edits (deletion, insertion, substitution) – and how induction impacts this efficiency over time - a crucial dataset for informing the scientific community about what can be achieved with the technology, and what efficiency to expect. Similarly, Tong et al precisely investigated the optimal PEgRNA design parameters (PBS and RTT length) to allow maximal usability of their method. The authors demonstrate the use of their technology both for editing plasmid or chromosomal DNA targets, and they reveal an extremely low off-target mutations in this process.

In addition, the authors attempted to increase efficiency of the technique through nicking the complementary strand, a reasonable expectation given that it works in eukaryotic cells, and report that this strategy does not work in bacteria.

Although the manuscript could gain from a better comparison of the method with the most recent literature on genome editing techniques, it certainly describes a useful tool for microbiologists, with potential for various applications and transferability to other bacteria than *E. coli*.

General comments

Line 99: Why giving the technique a new name? It feels like CRISPRnRAGE could simply be called Prime Editing in *E. coli* cells. Indeed, the proximity between the original Prime Editing method from Anzalone et al. and its implementation here in *E. coli* is such that it does not deserve to be called differently, even if this implementation in bacteria was not a trivial achievement. What's more, the CRISPRnRAGE acronym does not mention 'bacteria' or '*E. coli*', which really feels like re-naming Prime Editing differently, while the 'PEgRNA' ("Prime Editing gRNA") nomenclature, directly stemming from Prime Editing, is maintained.

Keeping the name Prime Editing would not diminish the interest of this work in my opinion, it would just clarify its filiation to the Anzalone method.

Line 52 to 62; line 176; and in discussion line 280 to 284: Only the initial MAGE technique is discussed in the text, and its later and more recent developments are not mentioned. For example, the pORTMAGE system from Wannier et al. reports editing efficiencies of 50%, and the system is shown to work in different bacteria.

Wannier, T. M. et al. Improved bacterial recombineering by parallelized protein discovery. *Proc National Acad Sci* 117, 13689–13698 (2020).

In more recent developments the efficiency was pushed even higher:

Schubert, M. G. et al. High throughput functional variant screens via in-vivo production of single-stranded DNA. *Biorxiv* 2020.03.05.975441 (2020) doi:10.1101/2020.03.05.975441.

The results obtained with CRISPRnRAGE should be put in perspective of these recent technologies. Of note, off-targets in the pORTMAGE and other recombineering systems is much higher, probably because it requires turning off the mismatch-repair system in *E. coli*, and they also require deletions in the host strain to reach full editing efficiency potential, such as *sbcB* or *recJ*.

Minor comments

Line 49: replace 'are' by 'use'

Line 142 to 152: Deep sequencing of the edited population – as it was done to analyze off-target mutations later in the manuscript - would have been informative here to thoroughly evaluate the extent of edit incorporation in the population, and how its frequency increases after longer induction time. Indeed, none of the edit types in Figure 2 shows complete editing efficiency across the 24 non-fluorescent colonies that were picked. Sequencing of the entire population (after selection in liquid) would have offered a more precise count of the actual editing efficiency. (This would have been useful, but it is not essential. Sanger sequencing, and later on deep sequencing of isolated colonies already unambiguously confirm the different edit incorporation).

Figure 1f: The plate views need to be zoomed in. It is very hard to distinguish fluorescence VS non-fluorescent colonies. Showing less colonies, but bigger, would make it easier.

Figure 2: The color-coded sequence is really useful to understand the associated PEGRNA constructs. A detail though, the blue highlight does not truly shows the RTT, rather the sequence to be replaced by the RTT that will contain the different edits listed.

In supplements

Line 100, 102: replace ul by μ l

Reviewer #2:

Remarks to the Author:

In this work, Tong and coworkers report the successful implementation of prime editors in *Escherichia coli*. They created a multi-plasmid system called CRISPR-nRAGE and show that it could introduce ranging small insertions, deletions, and substitutions in a GFP reporter encoded on a plasmid. Cells lacking GFP fluorescence generally contained the desired edit, although one edit (a 1-bp substitution) resulted in a higher frequency of other edits around the nicking site. They also showed that dual editing could be achieved, albeit with <1% editing, while introducing a second nick on the non-target strand resulted in extensive cytotoxicity. Building on successfully editing single sites, the authors explored different parameters affecting plasmid editing, including induction time and strength, length of the primer binding site, and length of the reverse transcription template. They also found that editing did not come with any appreciable off-target editing. Finally, the authors showed that two different chromosomal genes (*lacZ*, *galK*) could be edited. Based on these results, the authors conclude that CRISPR-nRAGE could be a useful tool for genome editing in *E. coli* and other organisms.

CRISPR-based tools have continued to advance in bacteria, although current approaches still rely on Cas9 that merely counter-selects against cells that did not undergo recombineering or on base editors that can introduce an extremely narrow range of edits. The recent advent of prime editing offers a distinct opportunity by driving editing and expanding the editing range of base editors. To date though, prime editors have only been implemented in eukaryotic cells. This work provides the first demonstration of prime editing in bacteria and indicates how it can be used to create different types of edits. This is a notable achievement that could lead to broader use of prime editing in bacteria, although the authors could do more beyond showing that what worked in eukaryotes also worked in *Escherichia coli* (with the exception of dual nicking) for two chromosomal genes. More details can be found below. Otherwise, the text is clearly written, the figures are generally well composed, and the data support the authors' conclusions.

Major comments:

1. The final demonstration of CRISPR-nRAGE involved editing two chromosomal genes in *E. coli*. Further extension would be helpful to fully convince others of the utility of prime editing in bacteria. This could include implementing prime editing in other bacteria beyond *E. coli*, demonstrating a larger set of edits across the genome, or working toward some application (e.g. performing site-directed mutagenesis at a target locus). The authors do provide more extensive data for plasmid editing, although this is considered easier than chromosomal editing, as seen for

base editing.

2. The authors consistently focus on cells lacking the functional protein, yet there are always cells with the functional. Some interrogation of these cells would help reveal why editing did not occur. For instance, for counter-selection with Cas9, these escapers normally possess an inactivated CRISPR component that prevented successful targeting. Prime editing could result in something different, and elucidating escape would help suggest how the editing frequency could be enhanced.

3. One common issue in bacteria is that editors prove cytotoxic, whether based on their mechanism of action (e.g. counter-selection) or the over-expression of different domains (e.g. cytidine deaminase for base editors). From the presented data though, there is no way to gauge if there was any loss in fitness from expressing the prime editor or from DNA targeting. This can be easily determined by measuring CFU's following induction, with appropriate controls (e.g. a non-targeting sgRNA).

Minor comments:

4. One emerging tool not addressed by the authors is the CRISPR transposon. While there are notable distinctions between CRISPR transposons and prime editors, these other tools should at least be addressed as part of the CRISPR toolbox for genome editing in bacteria.

5. L. 42: change "organisms" to "bacteria", as CRISPR-nRAGE is really geared to bacteria.

6. L. 56-59: MAGE doesn't require inactivation of MMR. Instead, eliminating this pathway boosts editing for certain types of small edits.

7. L. 75-77: Repair principally takes place with sister chromatids. A supplied repair can also be incorporated, leading to successful recombineering (see PMID = 27060147).

8. L. 96: If 1d is cited first, then this should be 1a. Alternatively, don't cite anything or provide a separate diagram generally depicting prime editors.

9. L. 113-114: I assume the authors are using SpyCas9. Explicitly state so here.

10. L. 133-135: technically, the depicted repair pathway is only hypothesized and has never been confirmed in vivo.

11. L. 150-151: provide the Sanger sequencing results, as there are many reasons why an outgrown colony could lose GFP fluorescence.

12. L. 156 (and Figure 2 legend): Use a different phrase besides off-targeting, as all of these edits are still associated with standard recognition of the sgRNA target. I would instead consider these unintended edits to the target.

13. L. 182-183: Briefly elaborate on why the PBS is limited to the length of the sgRNA target. Also, the PBS:non-target DNA strand R-loop theoretically could extend beyond the guide:target DNA strand R-loop.

14. L. 200: provide the Sanger sequencing results to show the desired insertions did in fact occur.

15. L. 204-205: did any of the resulting colonies have the desired edit (or at least show the color change)?

16. L. 205-206: this statement is shaky, as having NHEJ wouldn't necessarily allow the second nicking event to drive efficient prime editing. Also, this example involves plasmid editing, which may undergo repair differently than the chromosome.

17. L. 208: choose a different word than "viability", as this initially implies the viability of the E. coli.

18. L. 257: can you say more about this 1-nt substitution? Is there anything to suggest this site might be a true off-target?

19. L.269-270: this statement is shaky, as the viral-derived recombinases are being used in place of the endogenous machinery.

20. L. 283-284: this statement is somewhat misleading, as oligo-mediated recombineering still works quite well with lambdaRED and Cas9 counter-selection, and the cited paper substantially boosted the editing frequency with a small tweak to the sgRNA guide.

21. L. 300: this statement should incorporate the fact that the one attempt at multiplexing in this work was extremely inefficient.

22. Editing data in all figures and related main text: the authors report the editing efficiency, although not all colonies lacking the reporter protein contained the desired edit. To better align the axis label (and associated text) with what is being measured, these should be changed to reflect the observation rather than the underlying mechanism (e.g. to GFP-negative cells).

23. Figure 1F: I found these images too small to make out. Even when blown up, it's also not obvious there's a difference between aTc- and aTc+.

24. Figure 2C: how far apart are these two sites? This should be incorporated into the figure.

25. Figure 4: the streaks in b appear virtually all white, whereas c shows that a majority of the cells were not edited. This strikes me as a notable discrepancy, even if the sequencing results confirm the edits. Also, editing of galK should be quantified, such as by comparing the number of colonies with or without the inhibitor.

A point-by-point response to the reviewers' comments

We would like to thank the reviewers for their valuable comments and suggestions to improve our manuscript. Please find our point-by-point responses below in blue.

REVIEWER COMMENTS

Reviewer #1 (Remarks to the Author):

Tong et al report the adaptation of a remarkable technology called Prime Editing, for the first time in bacterial cells. The transfer of such a complex genetic editing platform in bacteria was not trivial, given that it requires the expression of a massive ~238 kDa fusion protein, and it relies on the cell endogenous DNA repair systems to incorporate the edit. The system developed by Tong et al, named CRISPRnRAGE, is surprisingly efficient, and represents a useful addition to the precise genetic editing toolkit in *E. coli*.

In particular, Tong et al. thoroughly investigated how the CRISPRnRAGE technique performs for different types of edits (deletion, insertion, substitution) – and how induction impacts this efficiency over time - a crucial dataset for informing the scientific community about what can be achieved with the technology, and what efficiency to expect. Similarly, Tong et al precisely investigated the optimal PEGRNA design parameters (PBS and RTT length) to allow maximal usability of their method. The authors demonstrate the use of their technology both for editing plasmid or chromosomal DNA targets, and they reveal an extremely low off-target mutations in this process.

In addition, the authors attempted to increase efficiency of the technique through nicking the complementary strand, a reasonable expectation given that it works in eukaryotic cells, and report that this strategy does not work in bacteria.

Although the manuscript could gain from a better comparison of the method with the most recent literature on genome editing techniques, it certainly describes a useful tool for microbiologists, with potential for various applications and transferability to other bacteria than *E. coli*.

Response: We highly appreciate these positive comments.

General comments

Line 99: Why giving the technique a new name? It feels like CRISPRnRAGE could simply be called Prime Editing in *E. coli* cells. Indeed, the proximity between the original Prime Editing method from Anzalone et al. and its implementation here in *E. coli* is such that it does not deserve to be called differently, even if this implementation in bacteria was not a trivial achievement. What's more, the CRISPRnRAGE acronym does not mention 'bacteria' or '*E. coli*', which really feels like re-naming Prime Editing differently, while the 'PEgRNA' ("Prime Editing gRNA") nomenclature, directly stemming from Prime Editing, is maintained.

Keeping the name Prime Editing would not diminish the interest of this work in my opinion, it would just clarify its filiation to the Anzalone method.

Response: Initially, for branding purpose, we wanted to name the CRISPR tools developed in the lab with "CRISPR-XXX", for example, we named the tailored base editor for streptomyces "CRISPR-BEST (CRISPR-Base Editing SysTem)", Tong, et al. *Proc. Natl. Acad. Sci. U.S.A.*, 116, 20366-20375.

But we totally agree with the reviewer, therefore the name CRISPR-nRAGE was replaced by CRISPR-Prime Editing for *E. coli* in the revision, necessary changes were made accordingly through the manuscript.

Line 52 to 62; line 176; and in discussion line 280 to 284: Only the initial MAGE technique is discussed in the text, and its later and more recent developments are not mentioned. For example, the pORTMAGE system from Wannier et al. reports editing efficiencies of 50%, and the system is shown to work in different bacteria.

Wannier, T. M. et al. Improved bacterial recombineering by parallelized protein discovery. *Proc National Acad Sci* 117, 13689–13698 (2020).

In more recent developments the efficiency was pushed even higher:

Schubert, M. G. et al. High throughput functional variant screens via in-vivo production of single-stranded DNA. *Biorxiv* 2020.03.05.975441 (2020) doi:10.1101/2020.03.05.975441.

The results obtained with CRISPRnRAGE should be put in perspective of these recent technologies. Of note, off-targets in the pORTMAGE and other recombineering systems is much higher, probably because it requires turning off the mismatch-repair system in *E. coli*,

and they also require deletions in the host strain to reach full editing efficiency potential, such as sbcB or recJ.

Response: As you suggested, we added further examples in the recent development of MAGE, including the pORTMAGE systems, and the retron based recombineering method.

The paragraph now reads:

Classical MAGE not only requires the synthesis and delivery of ssDNA oligos but also the expression of lambda (λ) red recombinase systems (Exo, Beta and Gam) in the target E. coli strain³. Several improved methods have been developed based on the classical MAGE to increase the editing efficiency and decrease the off-target effect. For example, the pORTMAGE system⁴, using a dominant-negative mutant protein of the MMR pathway, not only achieves higher editing efficiency and lower off-target effect, but also works for different bacterial species other than E. coli. One step forward, an improved pORTMAGE system was built by discovery of new, highly active single-stranded DNA-annealing proteins (SSAP). The identified CspRec improved pORTMAGE editing efficiency to up to 50%⁵. Recently, retron library recombineering was introduced as a new method that achieves up to 90% editing efficiency by in vivo production of single-stranded DNA using the targeted reverse-transcription activity of retrons⁶

Furthermore, we added a perspective to the Discussion in the revision as suggested, after "...E. coli without requiring DSBs, editing templates, or homologous recombination": *"We observed a very high fidelity of using the system in E. coli, while the mutation rates/off-target effects in the MAGE and other recombineering systems are much higher, and normally it requires pre-engineering of the host strains when using these systems³⁻⁵."*

Minor comments

Line 49: replace 'are' by 'use'

Response: Done.

Line 142 to 152: Deep sequencing of the edited population – as it was done to analyze off-target mutations later in the manuscript - would have been informative here to thoroughly

evaluate the extent of edit incorporation in the population, and how its frequency increases after longer induction time. Indeed, none of the edit types in Figure 2 shows complete editing efficiency across the 24 non-fluorescent colonies that were picked. Sequencing of the entire population (after selection in liquid) would have offered a more precise count of the actual editing efficiency. (This would have been useful, but it is not essential. Sanger sequencing, and later on deep sequencing of isolated colonies already unambiguously confirm the different edit incorporation).

Response: We agree with the reviewer that the proposed deep sequencing approach could provide additional information on the whole population level. As the reviewer stated, we have thoroughly characterized the various edits using an alternative approach. Therefore, we felt, as also mentioned by the reviewer, that the additional knowledge obtained from a population level analysis may not justify the significant experimental efforts, time requirements and costs and propose to keep the text unchanged.

Figure 1f: The plate views need to be zoomed in. It is very hard to distinguish fluorescence VS non-fluorescent colonies. Showing less colonies, but bigger, would make it easier.

Response: The figure was replaced with a zoomed in variant as suggested.

Figure 2: The color-coded sequence is really useful to understand the associated PEgRNA constructs. A detail though, the blue highlight does not truly shows the RTT, rather the sequence to be replaced by the RTT that will contain the different edits listed.

Response: We thank the reviewer for point this out, the blue highlight in the GFP reference sequence indeed shows the RTT, while the cyan masked Sanger sequencing traces show the sequence to be replaced by the RTT that will contain the different edits listed. To make it clear, we added “The cyan masked Sanger sequencing traces show the sequence to be replaced by the RTT that will contain the different edits designed” in the figure legend of Figure 2.

In supplements

Line 100, 102: replace ul by μ l

Response: Done.

Reviewer #2 (Remarks to the Author):

In this work, Tong and coworkers report the successful implementation of prime editors in *Escherichia coli*. They created a multi-plasmid system called CRISPR-nRAGE and show that it could introduce ranging small insertions, deletions, and substitutions in a GFP reporter encoded on a plasmid. Cells lacking GFP fluorescence generally contained the desired edit, although one edit (a 1-bp substitution) resulted in a higher frequency of other edits around the nicking site. They also showed that dual editing could be achieved, albeit with <1% editing, while introducing a second nick on the non-target strand resulted in extensive cytotoxicity. Building on successfully editing single sites, the authors explored different parameters affecting plasmid editing, including induction time and strength, length of the primer binding site, and length of the reverse transcription template. They also found that editing did not come with any appreciable off-target editing. Finally, the authors showed that two different chromosomal genes (*lacZ*, *galk*) could be edited. Based on these results, the authors conclude that CRISPR-nRAGE could be a useful tool for genome editing in *E. coli* and other organisms.

CRISPR-based tools have continued to advance in bacteria, although current approaches still rely on Cas9 that merely counter-selects against cells that did not undergo recombineering or on base editors that can introduce an extremely narrow range of edits. The recent advent of prime editing offers a distinct opportunity by driving editing and expanding the editing range of base editors. To date though, prime editors have only been implemented in eukaryotic cells. This work provides the first demonstration of prime editing in bacteria and indicates how it can be used to create different types of edits. This is a notable achievement that could lead to broader use of prime editing in bacteria, although the authors could do more beyond showing that what worked in eukaryotes also worked in *Escherichia coli* (with the exception of dual nicking) for two chromosomal genes. More details can be found below. Otherwise, the text is clearly written, the figures are generally well composed, and the data support the authors' conclusions.

Response: We thank the reviewer very much for these positive comments and praises, which

encourages us to do more following independent studies. All your other concerns are answered/addressed below.

Major comments:

1. The final demonstration of CRISPR-nRAGE involved editing two chromosomal genes in *E. coli*. Further extension would be helpful to fully convince others of the utility of prime editing in bacteria. This could include implementing prime editing in other bacteria beyond *E. coli*, demonstrating a larger set of edits across the genome, or working toward some application (e.g. performing site-directed mutagenesis at a target locus). The authors do provide more extensive data for plasmid editing, although this is considered easier than chromosomal editing, as seen for base editing.

Response: We agree with the reviewer that there is a high potential of CRISPR-Prime Editing in prokaryotes beyond *E. coli*. However, in this study, the main purpose is to demonstrate the versatility of CRISPR-Prime Editing in *E. coli*, which we feel is already a “round story” with appropriate contents. We currently are indeed trying to optimize the system and also trying to push it to other bacteria other than *E. coli*. We plan to publish these systems as follow-up studies as soon as all required (control) experiments have been carried out – which often unfortunately is a quite lengthy task on non-model organisms.

2. The authors consistently focus on cells lacking the functional protein, yet there are always cells with the functional. Some interrogation of these cells would help reveal why editing did not occur. For instance, for counter-selection with Cas9, these escapers normally possess an inactivated CRISPR component that prevented successful targeting. Prime editing could result in something different, and elucidating escape would help suggest how the editing frequency could be enhanced.

Response: The reviewer asked a very important question on how to better understand and use CRISPR-Prime Editing systems. For investigating how the escaper prevented editing by the CRISPR-Prime Editing system, we whole genome-sequenced 10 clones carrying an active CRISPR-Prime Editing system yet still have the GFP (the target) expressed. Interestingly, we found 7 out of the 10 escapers have lost the 26-bp 3' extension to survive

from the CRISPR-Prime Editing system. While the other three escapers have both plasmids and chromosome intact, no interpretable mutations that would explain how they escape were found across the chromosome and plasmids, which suggest that there are further yet-unknown escaping mechanisms beyond the modification of guide RNAs. We added these data in the Result section of Characterization of CRISPR-Prime Editing system in *E. coli* in the revision as follow: *"In several of these cases, the editing efficiency was low. Many clones carrying the activated CRISPR-Prime Editing systems still showed GFP fluorescence. We randomly picked 10 of these "escapers", together with four controls (Supplementary Table 1). The 14 strains were sequenced, and analyzed with our genome-wide SNP profiling approach that was used for the off/on-target evaluation as well. 7 out of the 10 "escapers" lost the 26-bp 3 prime extension sequence (Supplementary Table 5); except these deletions, the other parts of plasmids and the chromosome were intact. In 3 "escapers", no mutations/SNPs were identified both on plasmids and chromosome that can explain why no CRISPR-Prime Editing occurred (Supplementary Table 5). This indicates that besides mutating the guide RNA, yet-unknown escaping mechanisms are also present in E. coli."*

The data set was displayed in the Supplementary Table 5, the Illumina sequencing data was deposited to Dryad.

3. One common issue in bacteria is that editors prove cytotoxic, whether based on their mechanism of action (e.g. counter-selection) or the over-expression of different domains (e.g. cytidine deaminase for base editors). From the presented data though, there is no way to gauge if there was any loss in fitness from expressing the prime editor or from DNA targeting. This can be easily determined by measuring CFU's following induction, with appropriate controls (e.g. a non-targeting sgRNA).

Response: As suggested, we carried out a CFU assay with *E. coli* having GFP plasmid only, nRAGE plasmid only, GFP plasmid + nRAGE plasmid, GFP plasmid + nRAGE plasmid + non-targeting sgRNA, and GFP plasmid + nRAGE plasmid + GFP targeting plasmid both under non-induction and induction conditions. The result is displayed in Response Table 1:

Response Table 1. A CFU assay of *E. coli* strains transformed with different plasmids

Plasmid	LB Plate (200 ng/ml of ATc, 100 ug/ml of Spec and Amp, and 25 ug/ml of Chl were used)	CFU	Induction/non-induction
pCDF	Spec	5391000	
pCDF	Spec + ATc	4293000	79.6%
pCDF + pCRISPR-nRAGE	Spec + Chl	11070	
pCDF + pCRISPR-nRAGE	Spec + Chl + ATc	9270	83.7%
pCDF + pCRISPR-nRAGE + GFP-del	Spec + Chl + Amp	1530	
pCDF + pCRISPR-nRAGE + GFP-del	Spec + Chl + Amp + ATc	810	52.9%
pCDF + pCRISPR-nRAGE + bgRNA	Spec + Chl + Amp	810	
pCDF + pCRISPR-nRAGE + bgRNA	Spec + Chl + Amp + ATc	405	50.0%

White background: not induced; grey background: induced with ATc

From these results, we can conclude:

- (1) 200 ng/ml of ATc clearly affects the fitness of *E. coli* in this study;
- (2) One-pot multiple plasmids transformation dramatically reduced the transformation efficiency.
- (3) DNA targeting does not cause any loss in fitness (please see the highlighted comparison, yellow vs. yellow; cyan vs. cyan).

Additionally, from our results on the whole genome-based off-target evaluation and escaping mechanism analysis, we did not observe many mutations, which would indirectly indicate that the expression of Prime Editing system and DNA targeting gRNA do not have obvious toxicity in *E. coli*.

Minor comments:

4. One emerging tool not addressed by the authors is the CRISPR transposon. While there are notable distinctions between CRISPR transposons and prime editors, these other tools should at least be addressed as part of the CRISPR toolbox for genome editing in bacteria.

Response: We added the CRISPR transposon tools in the introduction as follows:

For the insertion of large DNA fragments, methods such as CRISPR-associated transposase (CAST)¹⁸ and INsert Transposable Elements by Guide RNA-Assisted TargEting

(INTEGRATE)¹⁹ were developed by combining CRISPR-Cas systems and transposons. The INTEGRATE was successfully tested in E. coli for integrating a ~10.1 kb fragment into the chromosome¹⁹.

5. L. 42: change “organisms” to “bacteria”, as CRISPR-nRAGE is really geared to bacteria.

Response: Done.

6. L. 56-59: MAGE doesn't require inactivation of MMR. Instead, eliminating this pathway boosts editing for certain types of small edits.

Response: Thank you for pointing to this imprecision, we re-wrote this part (also addressing the related comments from Reviewer #1)., The revised paragraph reads: “*Classical MAGE not only requires the synthesis and delivery of ssDNA oligos but also the expression of lambda (λ) red recombinase systems (Exo, Beta and Gam) in the target E. coli strain³. Several improved methods have been developed based on the classical MAGE to increase the editing efficiency and decrease the off-target effect. For example, the pORTMAGE system⁴, using a dominant-negative mutant protein of the MMR pathway, not only achieves higher editing efficiency and lower off-target effect, but also works for different bacterial species other than E. coli. One step forward, an improved pORTMAGE system was built by discovery of new, highly active single-stranded DNA-annealing proteins (SSAP). The identified CspRec improved pORTMAGE editing efficiency to up to 50%⁵. Recently, retron library recombineering was introduced as a new method that achieves up to 90% editing efficiency by in vivo production of single-stranded DNA using the targeted reverse-transcription activity of retrons⁶.*”.

7. L. 75-77: Repair principally takes place with sister chromatids. A supplied repair can also be incorporated, leading to successful recombineering (see PMID = 27060147).

Response: We added this information as follows: “*In these organisms, DNA damage is primarily repaired via HDR with sister chromatids¹², where the template DNA replace the damaged DNA fragment by recombination¹³.*” in the revision after “In most bacteria, DSBs normally lead to cell death due to the lack of NHEJ¹¹.”.

8. L. 96: If 1d is cited first, then this should be 1a. Alternatively, don't cite anything or provide a separate diagram generally depicting prime editors.

Response: We removed this citation of Figure 1d in the revision to not cause confusion.

9. L. 113-114: I assume the authors are using SpyCas9. Explicitly state so here.

Response: Yes, it is SpyCas9, we changed Cas9 to SpyCas9 in the revision.

10. L. 133-135: technically, the depicted repair pathway is only hypothesized and has never been confirmed in vivo.

Response: We totally agree, we added the word “hypothetically” in the description as “*After the reverse transcription process, the nicked double stranded DNA hypothetically undergoes an equilibration between the edited 3' flap and the unedited 5' flap. The cleavage of the unedited 5' flap then leads to the desired DNA editing¹⁴ (Fig. 1e).*”

11. L. 150-151: provide the Sanger sequencing results, as there are many reasons why an outgrown colony could lose GFP fluorescence.

Response: Thank you for pointing this out, we displayed the Sanger sequencing traces of two outgrown colonies with GFP-1bp-del (from Supplementary Fig. 1a) and GFP-3bp-insertion (from Supplementary Fig. 1b) in Supplementary Fig. 1e.

12. L. 156 (and Figure 2 legend): Use a different phrase besides off-targeting, as all of these edits are still associated with standard recognition of the sgRNA target. I would instead consider these unintended edits to the target.

Response: We have modified the expression accordingly.

13. L. 182-183: Briefly elaborate on why the PBS is limited to the length of the sgRNA target. Also, the PBS:non-target DNA strand R-loop theoretically could extend beyond the guide:target DNA strand R-loop.

Response: We did this calculation based on the widely-accepted concept that the Cas9

nuclease cuts 3-nt upstream of the PAM site, and the R-loop is restricted to 20-nt in size if a 20-nt spacer is used to construct the sgRNA (Szczelkun, M. D. et al. Direct observation of R-loop formation by single RNA-guided Cas9 and Cascade effector complexes. *Proc Natl Acad Sci U S A*. 111; 27. (2014); Zhu, X., Clarke, R., Puppala, A.K. et al. Cryo-EM structures reveal coordinated domain motions that govern DNA cleavage by Cas9. *Nat Struct Mol Biol* 26, 679–685 (2019)), as the flanking non-targeted DNA is twisted, which would limit the R-loop extension. However, we agree that the R-loop formation and size have not been completely elucidated, and require additional studies. In order to make the statement scientifically sound, we removed this sentence “As a 20-nt spacer was used in the sgRNA construct, the maximum theoretical length of PBS is 17 nt.” in the revision.

14. L. 200: provide the Sanger sequencing results to show the desired insertions did in fact occur.

Response: We added a Supplementary Fig. 5 to show deletions and insertions. Of note, the 18 bp fragment designed is exactly a mini-T7 promoter.

15. L. 204-205: did any of the resulting colonies have the desired edit (or at least show the color change)?

Response: We thank the reviewer for this good question, please see Supplementary Note 1 and Supplementary Fig. 4, we indeed did not see any color changes nor desired editing by Sanger sequencing.

16. L. 205-206: this statement is shaky, as having NHEJ wouldn't necessarily allow the second nicking event to drive efficient prime editing. Also, this example involves plasmid editing, which may undergo repair differently than the chromosome.

Response: Thank the reviewer for this comment, indeed there are still not 100% clear how a second nick on the other strand can increase the editing efficiency in eukaryotes, we directly quote the description from the first publication reporting Prime Editing (*Nature* volume 576, pages149–157 (2019)) “...we previously used Cas9 nickase to nick the non-edited strand,

directing DNA repair to that strand using the edited strand as a template^{16,17,18}....”, which inspired us for carrying out this experiment.

For not causing confusion, we modified the text in the revision to “*Inspired by the observation that a second nick in the non-edited strand would increase the editing efficiency of CRISPR-Prime Editing in some mammalian cells¹⁷ and plant cells¹⁸, likely due to the DNA repair is directed to that strand using the edited strand as a template, we designed and validated two strategies of the second nick introduction in E. coli (Supplementary Note 1).*”

17. L. 208: choose a different word than “viability”, as this initially implies the viability of the E. coli.

Response: We change “viability” with “ability”.

18. L. 257: can you say more about this 1-nt substitution? Is there anything to suggest this site might be a true off-target?

Response: This is a very good question. The substitution is observed in both Sanger sequencing and Illumina sequencing. As the “A” deletion in the 5-bp upstream of the targeted site is still within the R-loop, we tend to believe that this is a direct effect (and thus off-target effect) that was caused by the Prime Editing system.

19. L.269-270: this statement is shaky, as the viral-derived recombinases are being used in place of the endogenous machinery.

Response: We thank the reviewer for raising this valuable concern, we agree with your comments, recombineering only uses phage recombinases instead of using the host RecA, both do not require DSBs. Therefore, we have removed the following sentences “*Like most prokaryotes, E. coli has no functional NHEJ pathway to repair fatal DNA damages like DSBs. Instead, E. coli primarily employs homologous recombination-based repair, which made the RecET and lambda red recombineering a major method of DNA engineering*” as it is not relevant for this manuscript, and replaced them with “*Widely used and versatile methods for genetic engineering of E. coli are RedET and lamda red-based recombineering^{1,2}, or MAGE-based approaches³⁻⁵.*” in the revision.

20. L. 283-284: this statement is somewhat misleading, as oligo-mediated recombineering still works quite well with lambdaRED and Cas9 counter-selection, and the cited paper substantially boosted the editing frequency with a small tweak to the sgRNA guide.

Response: We appreciate the carefulness of the reviewer, in order to avoid misleading, we removed the citation (Jiang, W., Bikard, D., Cox, D., Zhang, F. & Marraffini, L.A. RNA-guided editing of bacterial genomes using CRISPR-Cas systems. *Nat Biotechnol* 31, 233-239 (2013).)

21. L. 300: this statement should incorporate the fact that the one attempt at multiplexing in this work was extremely inefficient.

Response: In the revision, we added the following sentence for a clearer statement after "...applied for high-throughput mutagenesis applications.": "*However, it has to be noted that the editing efficiency was extremely low in our proof-of-concept multiplexing approach using the strategy of providing two PEgRNA delivery plasmids.*"

Additionally, we also added such statement in the abstract, now it reads: "*By providing a second guide RNA, CRISPR-Prime Editing for E. coli can be used for multiplexed editing with a relatively low efficiency.*", while the original version is "*By providing a second guide RNA, CRISPR-nRAGE can be used for multiplexed editing.*"

22. Editing data in all figures and related main text: the authors report the editing efficiency, although not all colonies lacking the reporter protein contained the desired edit. To better align the axis label (and associated text) with what is being measured, these should be changed to reflect the observation rather than the underlying mechanism (e.g. to GFP-negative cells).

Response: We agree with the reviewer. We modified the subsection heading of "Editing efficiency evaluation" in the Methods section to "*Editing efficiency evaluation using a fluorescence-based colony counting assay*". We also added information accordingly across the text.

Axis label of Figure 3 was updated to "*The ratio of GFP-negative clones/total clones*"; "*The ratio*

of the white clones (GFP-negative)/ total clones on a screening plate was used as the editing efficiencies.” was also added in the caption of Figure 3.

23. Figure 1F: I found these images too small to make out. Even when blown up, it’s also not obvious there’s a difference between aTc⁻ and aTc⁺.

Response: As reviewer #1 also mentioned this, in the revision, we have zoomed in the plate view to only focus on a small region.

24. Figure 2C: how far apart are these two sites? This should be incorporated into the figure.

Response: Two nick sites are 111 bp away, which was incorporated into Figure 2c as well as described accordingly in the caption.

25. Figure 4: the streaks in b appear virtually all white, whereas c shows that a majority of the cells were not edited. This strikes me as a notable discrepancy, even if the sequencing results confirm the edits. Also, editing of galK should be quantified, such as by comparing the number of colonies with or without the inhibitor.

Response: Sorry for the confusion, the content of plate view in Figure 4b was described by the draw next to the plate photo, the blue streak is the non-edited WT control; and the three white streaks are pure cultures of lacZ mutated strains with the GT to TA substitution, TAG insertion, and CG deletion. These clones are selected from a screening plate, then confirmed by Sanger sequencing, while Figure 4c was a color-based screening result, showing the ratio of non-blue / total colonies in an induction plate with X-gal. To make this clear, the caption of Figure 4b and 4c were updated accordingly in the revision as follow: *“b. Three clones of E. coli MG1655, where the inactivation of lacZ was confirmed by Sanger sequencing and a wild type E. coli MG1655 were re-streaked on an agar plate with X-gal. c. A bar chart shows the editing efficiencies of chromosomal DNA engineering by 3-bp insertion, 2-bp deletion and 2-bp substitution by calculating the ratio of white clones/total clones on an induction LB plate with X-gal supplemented.”*

Reviewers' Comments:

Reviewer #2:

Remarks to the Author:

The authors have taken multiple steps to address the reviewers' comments, including the inclusion of new experimental data. As part of the incorporated data, I was hoping to see some extension of the presented work—particularly beyond the two chromosomal targets, although I understand the extra time this work adds and the importance of publishing the first example of prime editing in bacteria.

There was one response that I felt was insufficient and needed further work—albeit work that should be quick and easy to complete. In response to Reviewer #2/Comment #3, the authors provide results from a single transformation to conclude that the prime editors themselves or DNA targeting do not cause any loss in fitness. For one, replicates are needed to demonstrate that these results are reproducible and hold up with statistical analyses. Second, CFU's wouldn't capture any effects that reduce growth but not viable cells. Instead, growing cells with the various constructs in liquid culture under inducing and non-inducing conditions would address this issue and possibly complement the CFU data.

The authors also note the introduction of retrons on p. 3 as part of the revisions. I recommend noting that the achieved editing efficiencies required disrupting multiple repair pathways in the cell. This is an important limitation that lends to the use of prime editing.

A point-by-point response to the reviewers' comments

We would like to thank you and the reviewer for their valuable comments and suggestions to improve our manuscript. Please find our point-by-point responses below in blue.

REVIEWERS' COMMENTS

Reviewer #2 (Remarks to the Author):

The authors have taken multiple steps to address the reviewers' comments, including the inclusion of new experimental data. As part of the incorporated data, I was hoping to see some extension of the presented work—particularly beyond the two chromosomal targets, although I understand the extra time this work adds and the importance of publishing the first example of prime editing in bacteria.

Response: We highly appreciate your understanding.

There was one response that I felt was insufficient and needed further work—albeit work that should be quick and easy to complete. In response to Reviewer #2/Comment #3, the authors provide results from a single transformation to conclude that the prime editors themselves or DNA targeting do not cause any loss in fitness. For one, replicates are needed to demonstrate that these results are reproducible and hold up with statistical analyses. Second, CFU's wouldn't capture any effects that reduce growth but not viable cells. Instead, growing cells with the various constructs in liquid culture under inducing and non-inducing conditions would address this issue and possibly complement the CFU data.

Response: We repeated the CFU experiments provided in the previous response with triplicates and included and discussed the data / additional method in the revised manuscript (line 145ff; methods were added line 482ff; Supplementary Table 5 was added to SI).

As suggested, we also carried out a 24h-liquid cultivation assay with and without induction of all constructs using a 96-well microtiter plate. Only little loss in fitness due to the prime editors or DNA targeting were observed.

The data was included in the manuscript (lines 148 ff, Supplementary Table 6, Supplementary Figure 6 were added to SI)

Response Figure 1 (Suppl. Figure 6). Growth profiles of *E. coli* strains transformed with different plasmids under induced and uninduced conditions.

Moreover, we also calculated the μ_{\max} and the maximum doubling time of strains bearing the same plasmids with or without induction based on the Monod equation. We could see that introducing additional plasmids indeed affect the growth of *E. coli*, moreover, inducer ATc at the concentration of 200 ng/ml also has certain negative effects on the growth of *E. coli*. (Suppl Table 6).

The authors also note the introduction of retrons on p. 3 as part of the revisions. I recommend noting that the achieved editing efficiencies required disrupting multiple repair pathways in the cell. This is an important limitation that lends to the use of prime editing.

Response: We clarified this as recommended by adding the following note to the statement (line 62-63) *..., however, such editing efficiencies require disrupting multiple DNA repair pathways in the host cell⁶, which heavily limits its applications*".